# Morbidity transition at the national and sub-national level and their determinants over the past and contemporary period in India

**Mahadev Bramhankar** *, **Murali Dhar**

Department of Bio-Statistics and Epidemiology, International Institute for Population Sciences (IIPS), Mumbai, India

* bramhankarakash@gmail.com

**Data Availability Statement:** Data used in the study was secondary micro data which has been directly retrieve from National Sample Survey Office (NSSO) official data. The NSSO data periodically collected in various rounds which has

## Abstract

The study delves into the epidemiological transition, examining the intricate changes in health status patterns and their connection to morbidities. Specifically, it assesses morbidity transition at both national and subnational levels in India and their determinants from 1995 to 2018. This study examines self-reported morbidities in India by utilising four rounds of National Sample Survey Organisation (NSSO) data (52nd, 60th, 71st, and 75th) from 1995–2018. We estimated prevalence by conducting descriptive analysis on socio-demographic determinants and morbidities such as: Infectious and Communicable Diseases (In&CDs), Non-communicable diseases (NCDs), Disability and other diseases. Moreover, we employed pooled regression analysis to explore morbidity risk transitions over the past decades. The study revealed a steady upsurge in morbidity prevalence in India, doubling from 56 (per thousand) in 1995 to 106 in 2014. However, a considerable decline was observed in the most recent round in 2018 (79 per thousand). From 1995 to 2018, NCDs gained a prominent share in morbidity trends. Kerala in the southern region reported the highest rates, followed by states like Lakshadweep, Andhra Pradesh, Karnataka, West Bengal, Punjab, and others. Age, sex, residence, education, caste, religion, and wealth are pivotal factors in determining the severity of different disease burdens in different sections of the population in India. Over the study period (1995, 2004, 2014, and 2018), the odds of reported morbidities risk transition significantly increased over successive decades: 1.81 times in 2004 (95% CI: 1.78–1.84), 2.16 times in 2014 (95% CI: 2.12–2.2), and 1.44 times in 2018 (95% CI: 1.41–1.46), compared to 1995 (52nd round). The study reveals significant disparities in morbidity reporting across Indian states from 1995 to 2018, attributed to distinct demographic, social, and economic determinants. India continues to grapple with the dual burden of diseases, but the NCDs burden is mounting at a faster pace than CDs.

been publicly available on their official site MOSPI National Data Archive (https://microdata.gov.in/nada43/index.php/catalog).

**Funding:** The author(s) received no specific funding for this work.

**Competing interests:** The authors have declared that no competing interests exist.

## Introduction

The epidemiological transition focuses on the complex change in patterns and trends of health and illness. It's interactions between patterns and their demographic, economic and sociologic determinants and consequences [1]. In the context of health, the 21st century is most likely to be "Asian countries" because of the shift in the disease burden and population composition, the process well underway at the close of the century. The beginning of this century provided an early preview of the health problems and challenges most countries will face in the coming decades, like emerging infectious diseases such as SARS-CoV-1, HIV-AIDS, Ebola, influenza, and Non-Communicable Diseases (NCDs). Recently, India has become the most populous country, with a population of over 1.42 billion across 28 states and 8 UTs. Many of India's major states have larger populations than most European and American countries [2,3]. Indian states have not only vast populations but also heterogeneity at the various levels of caste, religion, culture, geographical region, socio-economic factors, etc. India has more than two thousand ethnic groups and diverse lifestyles and it seems to be reflects a complex social fabric [4].

In India, morbidity patterns and causes of death have remarkable structural changes countrywide. After independence till 1980, the country was overwhelmed with the burden of infectious and parasitic diseases. However, since 1990, chronic diseases have dominated the burden of communicable diseases. An existing study brought to light that the mortality trend during 1969–1995 of infectious diseases declined from 47.7% to 22.1%, while Non-Communicable Diseases (NCDs) increased from 35.9% to 55% [5]. India's recent past phases of epidemiological transition resulted into high morbidity, low mortality and the double burden of communicable diseases and chronic diseases during the period of 1995 to 2004. During this period, the crude morbidity rate rose by more than 60% and 90% among rural and urban populations, respectively [6–8]. Another study showed a consistent upward trend in the prevalence of NCDs in general and Cardiovascular Diseases (CVDs) in particular throughout the past two decades [9]. In India, the epidemiological transition level may shift to the age of degenerative and man-made diseases from the age of receding pandemics [10]. According to a study of Epidemiological Transition Level (ETL) at the state level, there was found wide variation in ETL among the Indian states, ranging from the Highest in Kerala (0.16) to the lowest in Bihar (0.74).

In delving into the existing literature on the Global Burden of Disease (GBD) study and a limited number of peer studies, it becomes evident that a comprehensive exploration of India's epidemiological transition, specifically through morbidities, is notably lacking. The available studies often fall short due to incomplete data and a dearth of information on morbidities, particularly at the sub-national level and across diverse segments of the population. Unlike the GBD study and a handful of peer studies, our research aims to bridge this gap by focusing narrowly on India's epidemiological transition through morbidities, both at the national and state levels [11]. The scarcity of comprehensive and sub-national morbidity data has resulted in a lack of attention to the longitudinal aspects of the transition, hindering a nuanced understanding of the substantial shifts in morbidity and mortality over time [12,13].

Additionally, considering the impact of self-reported morbidities as a proxy measure for the population's health status [14]; our research aims to unravel the socio-demographic and economic determinants influencing these patterns. Therefore, our study seeks to contribute significantly to the understanding of India's epidemiological transition by addressing the gaps left by existing literature, utilizing comprehensive morbidity data, and understanding the socio-demographic factors that play a crucial role in shaping these patterns over the past three decadal period.

## Data and methods

We utilised secondary data from four rounds of National Sample Survey Organisation (NSSO) data; 52[nd] conducted in 1995–96, 60[th] in 2004, 71[st] in 2014–15, and 75[th] in 2018. During these survey rounds, a substantial number of households were surveyed: 120,942, 73,868, 65,932 and 113,823 households respectively. The survey's coverage extended to all states and union territories, employing a multi-stage stratified sampling methodology. The first-stage units were census villages in rural areas and urban frame survey blocks in urban areas. In the latest rounds, specifically the 75[th] and 71[st] NSS surveys, the theme was "India - Social Consumption: Health." The 60[th] round of NSS focused on "Morbidity and Health Care," while the 52[nd] round focused on the "Survey on Health Care." This consistent thematic approach enables valuable comparisons of health to be made across all NSS survey rounds. Detailed information regarding the survey's design and findings can be found in the respective reports for each round [7,15–17]. To conduct this study no ethical approval was required due to use of secondary data.

In each NSSO round, there were multiple kinds of diseases and disabilities collected; these self-reported morbidities have been classified into four categories: Infectious and communicable diseases (In&CDs), Non-communicable diseases (NCDs), Disability and Injury and other diseases. The classification of diseases was done according to the tenth International Classification of Diseases (ICD-10). As per the available information among different rounds, statistical analysis was carried out; the first study estimated the prevalence of self-reported morbidities for persons who had fallen ill during the span of the last fifteen days in India. The prevalence (Pi) of "i[th]" morbidities was estimated per 1000 population [9,16].

$$Pi = \frac{\textit{Number of morbid Persons due to ith Cause}}{\textit{Total number alive persons in the sample household}} * 1000$$

In the second stage, the study used bivariate analysis between socio-demographic characteristics and dependent variables (morbidities) such as IDs, NCDs, Disability and other diseases. The descriptive statistics of the self-reported disease conditions were estimated by socio-demographic determinants such as age, sex, place of residence, level of education, caste, religion, wealth, etc., in India. Moreover, the study employed multiple logistic and multiple multinomial logistic regressions for the pooled data of different NSS rounds. The following equation represents the logistic model,

$$\textbf{Logit}(\textbf{Y}) = \ln\left(\frac{\hat{p}}{1-\hat{p}}\right) = \boldsymbol{\beta_0} + \boldsymbol{\beta_1 x_1} + \cdots + \boldsymbol{\beta_n x_n}$$

Where, $\hat{p}$ is for the expected probability that the outcome is present; $x_1 \ldots x_n$ are distinct independent background variables in our study; and $\beta_1 \ldots \beta_n$ are the regression coefficients. This technique helps to understand the landscape of morbidity transition from an epidemiological perspective, encompassing changes in health status patterns over recent decades. The study applied this model to the pooled data from the four rounds of NSSO. We employed a pooled multiple and multinomial logistic regression model to assess the morbidity transition and their risk. It also seeks to explore and control the changes related to socio-demographic determinants and temporal shifts. We have used the STATA 16 software for the statistical analysis purposes. Regression results were provided as Adjusted Odds Ratio (AOR) from the pooled multiple logistic regression and Relative Risk Ratio (RRR) from the pooled multinomial logistic regression. Further, 95% Confidence interval and other parameter have been also provided for the essence of significance and variation.

## Results

### Prevalence of all and major types of self-reported morbidities

Fig 1 shows that the prevalence of all types of morbidities in India has been steadily increasing and doubled, i.e. from 56, 95 to 106 (per thousand populations) in 1995, 2004 and 2014, respectively. However, its trend has considerably declined to 79 per thousand in 2018. A similar kind of scenario has been observed among Infectious and Communicable Diseases (In&CDs), Non-communicable diseases (NCDs), Disability except Injury and other diseases started declining since 2004. Among different types of morbidities, NCDs have a major share of prevalence (30 per thousand in 2018), which has increased more than three times from (8.6 per thousand) in 1995. In&CDs also almost doubled from (8 to 15 per thousand) in the past two decades. Morbidities transition of NCDs, In&CDs and Injury and other diseases were noticeably high over two decades.

Fig 2 presents the state-level variation of all morbidities over the study period. Kerala consistently stands out with the highest reported morbidity rates, recording figures of 112 per 1000 population in 1995, 266 in 2004, 370 in 2014, and 294 in 2018. Following Kerala, states like Lakshadweep, Andhra Pradesh, West Bengal, Punjab, and others also exhibit noteworthy prevalence rates. Conversely, the northern and eastern states, particularly Manipur, exhibit the lowest prevalence of morbidity conditions. In 1995, Manipur reported a mere 7 cases per 1000 individuals, which increased to 28 in 2004, 29 in 2014, and declined to 19 in 2018. Mizoram, Nagaland, Meghalaya, and other northeastern states also follow a similar trend of relatively low morbidity rates. In the broader state-level context, the pattern of morbidity conditions appears to resemble a parabolic curve. It started with relatively low levels in 1995. Then, it

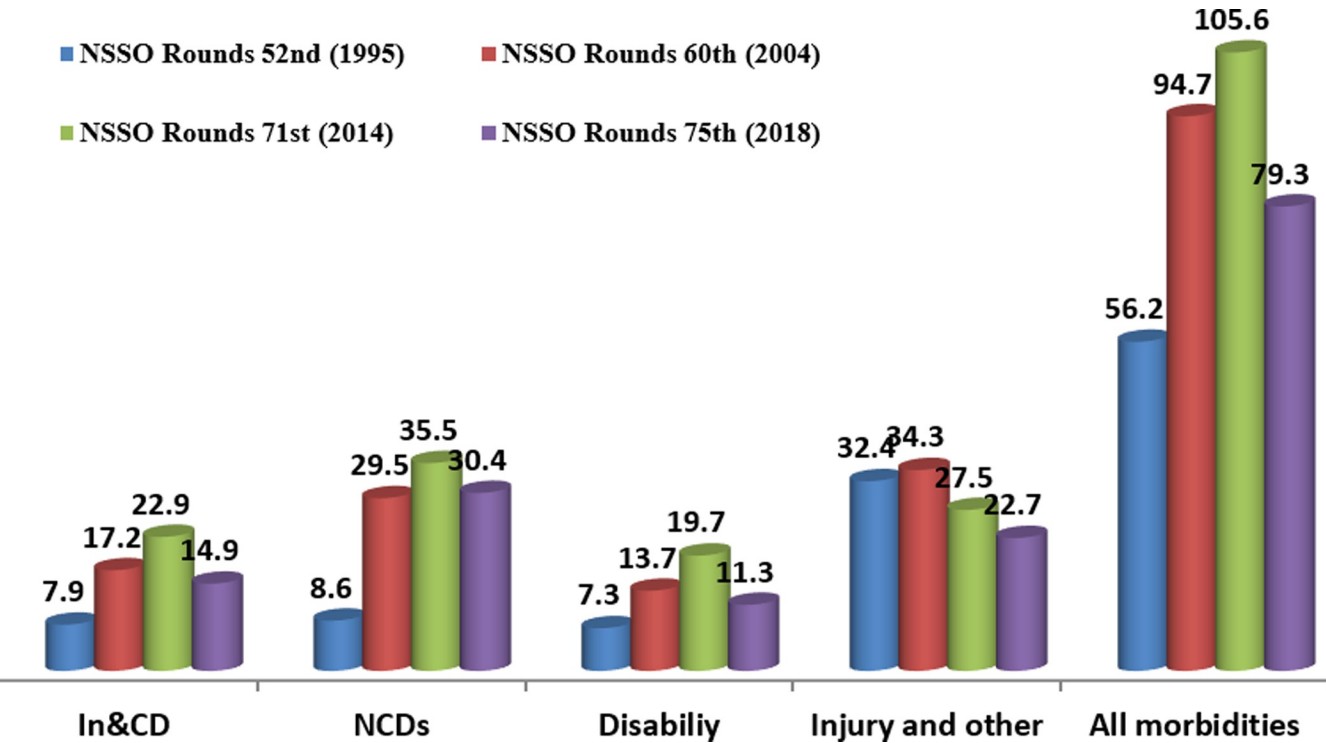

**Fig 1. Prevalence of self-reported morbidities (per 1000) in India, 1995–2018.**

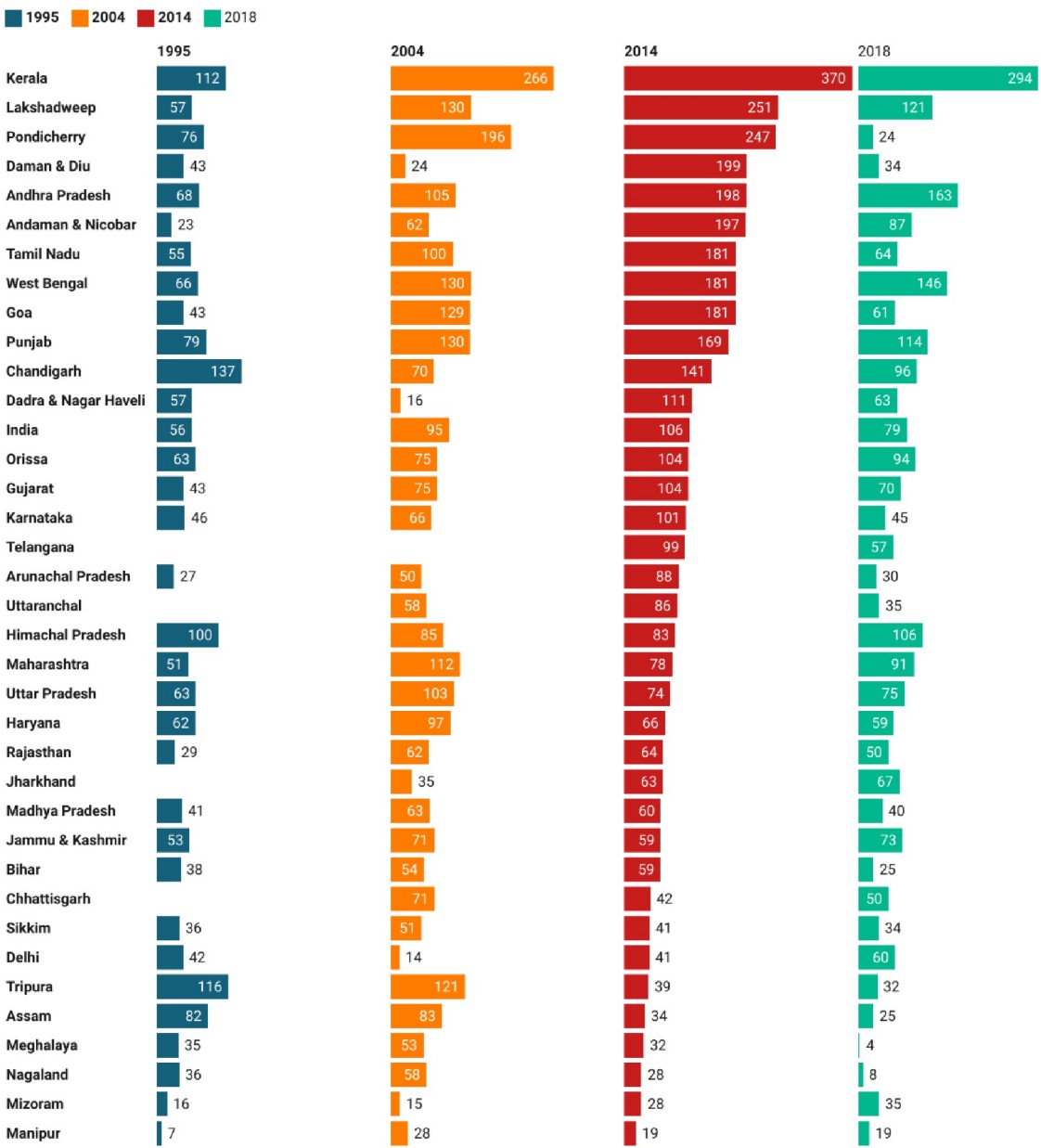

Prevalence is based on Self-Reported all kinds of morbidities reported during NSSO (52, 60, 71 and 75) rounds. Few states data points for 1995 and 2004 were unavailable because of the undivided at that time period from the parent state.
Chart: Authour • Created with Datawrapper

**Fig 2. Prevalence of self-reported all morbidities (per 1000) at the sub-national level in India, 1995–2018.**

exhibited an upward trend in 2004 and 2014 before gradually declining thereafter up to 2018, but it is still at a higher level as per recent prevalence.

Table 1 represents prevalence at the state level according to the major types of morbidities. Here, Infectious and Communicable Diseases (In & CDs) reported significant variations with wide ranges 4 to 80 per 1000 people reported across the Indian states throughout the period of 1995–2018. Also, there has been a considerable decline for less than half-a-decade in prevalence from 2014 to 2018 across all the states except Northern states such as J&K, Himachal

**Table 1. Prevalence of Major types of self-reported morbidities (per 1000) in India and its states, 1995–2018.**

| States/UTS | In&CD | | | | NCDs | | | | Disability | | | | Injury and other | | | |
|---|---|---|---|---|---|---|---|---|---|---|---|---|---|---|---|---|
| | 1995 | 2004 | 2014 | 2018 | 1995 | 2004 | 2014 | 2018 | 1995 | 2004 | 2014 | 2018 | 1995 | 2004 | 2014 | 2018 |
| Jammu & Kashmir | 8.4 | 11.1 | 14.3 | 32.4 | 9.7 | 27.9 | 17.4 | 21.8 | 7.0 | 12.9 | 19.7 | 7.0 | 27.7 | 18.6 | 7.8 | 11.6 |
| Himachal Pradesh | 13.7 | 14.0 | 16.7 | 19.3 | 16.1 | 30.8 | 26.9 | 43.3 | 21.0 | 15.8 | 24.5 | 14.6 | 48.8 | 24.4 | 14.8 | 28.6 |
| Punjab | 6.0 | 19.4 | 33.1 | 36.9 | 18.7 | 53.4 | 59.7 | 38.8 | 10.5 | 17.8 | 30.6 | 17.1 | 44.1 | 38.9 | 45.1 | 21.5 |
| Chandigarh | 2.6 | 3.8 | 14.6 | 6.5 | 19.0 | 30.6 | 43.5 | 57.7 | 8.8 | 2.2 | 34.7 | 15.7 | 106.5 | 33.8 | 48.6 | 16.2 |
| Uttaranchal | NA | 11.5 | 31.2 | 10.3 | NA | 11.6 | 9.5 | 6.9 | NA | 11.2 | 8.8 | 4.9 | NA | 23.5 | 36.3 | 13.0 |
| Haryana | 7.3 | 16.7 | 17.5 | 10.8 | 10.8 | 36.6 | 16.3 | 16.1 | 7.6 | 12.8 | 8.1 | 7.7 | 35.9 | 30.4 | 23.9 | 24.3 |
| Delhi | 5.4 | 0.9 | 13.8 | 10.5 | 8.3 | 4.9 | 7.1 | 17.0 | 6.2 | 1.9 | 4.3 | 5.8 | 22.1 | 6.3 | 15.4 | 27.0 |
| Rajasthan | 5.2 | 11.6 | 17.8 | 13.0 | 4.0 | 18.8 | 18.5 | 13.0 | 2.1 | 8.1 | 10.2 | 8.6 | 18.0 | 23.5 | 17.0 | 14.9 |
| Uttar Pradesh | 12.7 | 25.7 | 20.2 | 17.9 | 8.5 | 22.9 | 14.9 | 13.5 | 5.2 | 11.6 | 15.5 | 9.2 | 36.9 | 42.4 | 23.5 | 34.4 |
| Bihar | 6.0 | 14.5 | 20.6 | 8.2 | 5.9 | 11.0 | 8.7 | 2.8 | 6.5 | 4.7 | 11.4 | 3.5 | 19.3 | 23.3 | 18.2 | 10.6 |
| Sikkim | 3.6 | 9.5 | 8.1 | 5.7 | 6.1 | 23.2 | 7.0 | 14.1 | 3.2 | 9.5 | 13.4 | 6.1 | 23.1 | 8.5 | 12.5 | 7.9 |
| Arunachal Pradesh | 2.7 | 31.2 | 32.8 | 18.2 | 3.9 | 4.7 | 10.5 | 2.3 | 1.6 | 4.6 | 10.9 | 3.1 | 18.9 | 9.9 | 33.4 | 6.0 |
| Nagaland | 6.2 | 31.4 | 7.1 | 2.0 | 6.5 | 1.7 | 0.1 | 2.4 | 3.6 | 2.7 | 3.8 | 2.0 | 19.4 | 21.8 | 17.2 | 1.8 |
| Manipur | 0.4 | 6.3 | 9.3 | 5.5 | 2.8 | 6.1 | 3.6 | 2.9 | 0.4 | 1.9 | 2.2 | 2.5 | 3.7 | 13.6 | 3.9 | 7.8 |
| Mizoram | 0.2 | 6.5 | 5.9 | 10.3 | 5.1 | 2.1 | 6.7 | 9.0 | 1.3 | 2.3 | 6.1 | 6.3 | 9.4 | 4.0 | 9.5 | 8.9 |
| Tripura | 21.3 | 36.6 | 9.3 | 7.9 | 9.5 | 58.3 | 5.7 | 6.4 | 8.3 | 3.7 | 5.7 | 2.0 | 76.5 | 21.9 | 18.5 | 15.2 |
| Meghalaya | 10.2 | 23.1 | 16.7 | 1.7 | 1.7 | 2.0 | 1.4 | 0.2 | 4.6 | 12.7 | 4.4 | 1.1 | 18.0 | 15.0 | 9.3 | 0.6 |
| Assam | 21.3 | 28.0 | 10.8 | 9.4 | 8.2 | 13.2 | 3.1 | 5.0 | 6.0 | 8.6 | 6.4 | 4.2 | 46.9 | 33.2 | 13.2 | 6.4 |
| West Bengal | 11.2 | 26.3 | 36.4 | 19.0 | 13.0 | 41.9 | 51.2 | 66.7 | 7.9 | 17.0 | 39.8 | 24.1 | 33.8 | 45.0 | 53.4 | 36.1 |
| Jharkhand | NA | 7.5 | 14.5 | 20.8 | NA | 8.4 | 15.5 | 10.3 | NA | 1.8 | 12.8 | 8.5 | NA | 16.8 | 20.1 | 27.8 |
| Orissa | 6.0 | 18.6 | 25.4 | 19.5 | 4.1 | 11.9 | 18.2 | 23.6 | 4.1 | 7.0 | 19.5 | 15.2 | 48.6 | 37.6 | 41.1 | 35.7 |
| Chhattisgarh | NA | 11.8 | 17.7 | 14.9 | NA | 16.7 | 8.8 | 9.9 | NA | 9.1 | 4.6 | 3.6 | NA | 33.5 | 11.2 | 21.3 |
| Madhya Pradesh | 4.8 | 14.8 | 15.9 | 10.2 | 3.4 | 13.9 | 12.8 | 11.8 | 3.7 | 9.0 | 8.2 | 5.4 | 28.6 | 25.3 | 22.7 | 12.5 |
| Gujarat | 8.1 | 16.9 | 34.3 | 18.0 | 7.5 | 26.8 | 37.1 | 29.9 | 4.5 | 11.7 | 19.0 | 6.4 | 22.6 | 19.2 | 13.1 | 15.4 |
| Daman & Diu | 21.7 | 5.5 | 30.5 | 0.4 | 6.4 | 12.2 | 122.2 | 22.5 | 5.3 | 0.9 | 25.4 | 1.1 | 9.6 | 4.9 | 20.9 | 9.6 |
| Dadra & Nagar Haveli | 9.0 | 2.1 | 10.9 | 0.4 | 1.1 | 8.5 | 40.7 | 1.6 | 11.8 | 0.1 | 40.6 | 0.8 | 35.5 | 5.6 | 18.6 | 60.3 |
| Maharashtra | 5.3 | 16.8 | 27.5 | 20.1 | 7.8 | 38.6 | 21.5 | 37.3 | 8.5 | 23.2 | 13.4 | 10.0 | 29.7 | 33.6 | 15.3 | 23.3 |
| Andhra Pradesh | 5.2 | 9.4 | 20.2 | 15.0 | 9.6 | 39.4 | 100.1 | 89.2 | 15.6 | 20.3 | 40.7 | 33.4 | 37.1 | 35.8 | 37.1 | 25.6 |
| Karnataka | 5.6 | 7.7 | 19.1 | 7.0 | 7.9 | 23.2 | 37.3 | 19.1 | 8.0 | 12.3 | 19.4 | 6.1 | 24.0 | 23.2 | 25.1 | 12.7 |
| Goa | 2.6 | 11.0 | 80.1 | 1.5 | 8.6 | 92.3 | 83.1 | 54.3 | 10.5 | 6.4 | 2.2 | 2.6 | 20.9 | 19.4 | 15.1 | 2.3 |
| Lakshadweep | 2.5 | 11.1 | 59.2 | 9.2 | 22.6 | 61.9 | 106.0 | 78.6 | 4.8 | 21.2 | 47.5 | 27.2 | 26.7 | 35.7 | 37.8 | 5.6 |
| Kerala | 8.2 | 21.0 | 39.4 | 24.7 | 24.2 | 117.9 | 184.7 | 176.6 | 16.4 | 38.1 | 64.8 | 46.8 | 62.9 | 88.6 | 81.3 | 45.6 |
| Tamil Nadu | 6.3 | 13.2 | 25.3 | 5.1 | 9.8 | 34.2 | 90.0 | 38.4 | 6.3 | 15.6 | 28.7 | 6.6 | 32.1 | 37.1 | 37.0 | 14.1 |
| Pondicherry | 6.5 | 22.1 | 43.7 | 1.7 | 3.3 | 63.2 | 130.4 | 14.8 | 13.2 | 50.7 | 29.6 | 2.5 | 52.5 | 60.2 | 43.3 | 5.3 |
| Andaman & Nicobar | 4.5 | 8.8 | 70.6 | 12.8 | 1.2 | 22.5 | 58.1 | 60.9 | 1.2 | 8.5 | 40.6 | 7.9 | 16.5 | 22.0 | 27.5 | 5.5 |
| Telangana | NA | NA | 9.8 | 6.1 | NA | NA | 38.2 | 25.5 | NA | NA | 19.3 | 8.4 | NA | NA | 31.8 | 17.2 |
| **India** | **7.9** | **17.2** | **22.9** | **14.9** | **8.6** | **29.5** | **35.5** | **30.4** | **7.3** | **13.7** | **19.7** | **11.3** | **32.4** | **34.3** | **27.5** | **22.7** |

Pradesh, Punjab and Chandigarh. Still, there are a substantial number of states, namely, Kerala, West Bengal, Gujarat, Punjab, Maharashtra, Orissa, Uttar Pradesh, Himachal Pradesh, and J&K with the prevalence of In&CDs more than the national average of 15 per 1000.

In particular, the NCD profile among the population from 1995 to 2018 across the Indian states has revealed vast variation. The highest prevalence of NCDs has been reported in Kerala (118 in 2004, 185 in 2014, and 177 in 2018 per thousand individuals), followed by Pondicherry, Lakshadweep, Andhra Pradesh, Goa, Tamil Nadu, and Punjab. However, the lowest NCDs were reported in north-eastern states such as Meghalaya (1.7 in 1995, 2 in 2004, 1.4 in 2014,

and 0.2 in 2018 per 1000 individuals) followed by Nagaland, Assam, Manipur, Mizoram, Tripura, Arunachal Pradesh and Bihar. As reported by NCDs, prevalence has declined in several states in less than half a decade (2014 to 2018), but it is still sizably higher in numbers.

Similarly, the state-level burden pattern of disability was reported in parabolic over three decades. It was reported lower in 1995 and then a higher and increasing trend from 2004 to 2014, and then after a few years, it was reported lower than in 2018. Through the Disability pattern as like NCD burden, Kerala has the highest burden (16, 38, 65, and 47 in 1995, 2004, 2014 and 2018, respectively) reported among all states and UTs. Meanwhile, the lowest rates are in Manipur, Nagaland, Delhi, Goa, etc. The last fourth categorization of diseases, i.e., Injury and Other diseases, reported significantly higher numbers in 1995 and 2004 across almost all the states and then reported lower prevalence in 2014 and 2018. Again, Kerala reported an all-time high burden of injury and other diseases; it was 63 in 1995, and after two decades, it recently reported 46 per thousand in 2018. There were several other states, West Bengal, Chandigarh, Orissa, Andhra Pradesh, Haryana, Uttar Pradesh, Dadara & Nagar Haveli and Delhi, etc., have a higher burden of Injury and Other diseases than the national average (23 per thousand populations).

## Prevalence of all and major types of morbidities according to socio-demographic

Table 2 revealed that all morbidities have been significantly higher in people aged 45–59 and 60 and above compared to their counterparts and lowest among young, adolescent and working age categories (15–44). There has been a considerably higher prevalence among females at all times (59 in 1995, 101 in 2004, 119 in 2014 and 89 in 2018 per 1000) compared to men (53 in 1995, 89 in 2004, 93 in 2014 and 71 in 2018 per 1000). Education also plays a vital role in morbidities; the illiterate category has reported significantly higher morbidity than the other educational categories. As per place of residence, all morbidities were reported higher in the urban sector than in the rural; in a recent morbidity round (NSSO-75[th], 2018), it was reported 72 more than in the rural sector 98 per 1000 population in India. Wealth indicator MPCE is one of the essential economic determinants. It revealed that the prevalence of all morbidities increases as we shift from poorest to richest households for all four study periods (NSSO: 1995, 2004, 2014, and 2018). However, there was a slight decline in the burden of all morbidities from 2014 to 2018.

Table 3 represents major types of morbidity patterns that were observed almost similar to overall morbidity patterns according to the socio-demographic determinates. The In&CDs, NCDs, Disability and Injury & Other diseases prevalence has been increasing since 1995 to the highest in 2014, and it has been found to slightly decline in 2018 in the contemporary period across almost all socio-demographic determinates. Although the prevalence slightly declined, all types of morbidities were multiplied by their burden almost three times within less than thirty years (1995–2018). Among the older adults (60 and above) reported the highest infectious, NCDs, disability and Injuries, while the lowest was noted in 15–44 working-aged people in the past three decadal periods.

Among females, NCD burden (10 in 1995, 31 in 2004, 40 in 2014 and 34 in 2018 per 1000) was more prevalent than males (8 in 1995, 29 in 2004, 31 in 2014 and 27 in 2018 per 1000) but it has tripled the burden of NCDs in both male and females between 1995 to 2014. Marital status was also found to be one of the social determinates, which reveals that the In&CDs, NCDs, Disability and Injury were reported almost double and significantly higher among widowed, divorced or separated than the never-married and currently married people. In rural areas, In&CDs, and urban areas, NCDs were reported marginally higher than their counterparts.

**Table 2.  Prevalence of all morbidities (per 1000) according to socio-demographic determinates in India, 1995–2018.**

| Attributes | All morbidities | | | |
|---|---|---|---|---|
| | **1995** | **2004** | **2014** | **2018** |
| **Age Group (Years)** | | | | |
| 0–14 | 48.7 | 73.7 | 71.8 | 58.5 |
| 15–44 | 42.7 | 62.0 | 66.3 | 43.2 |
| 45–59 | 72.7 | 132.1 | 173.4 | 122.9 |
| 60 and above | 178.8 | 340.3 | 337.2 | 304.2 |
| **Sex** | | | | |
| Male | 53.9 | 88.9 | 92.8 | 70.5 |
| Female | 58.6 | 100.7 | 119.3 | 88.7 |
| **Education** | | | | |
| Illiterate | 63.8 | 112.2 | 129.5 | 106.8 |
| Literate and below primary | 50.3 | 76.9 | 95.6 | 75.7 |
| Primary and middle | 49 | 79.7 | 102.9 | 75.9 |
| Secondary and Higher Secondary | 44.1 | 84.5 | 90.1 | 64 |
| Graduation and above | 91.4 | 85.6 | 96.3 | 70.2 |
| other | 46.8 | 145.8 | 99.8 | 75.5 |
| **Marital status** | | | | |
| Never married | 45.3 | 66.6 | 63.5 | 49.8 |
| Currently married | 60.1 | 105.2 | 125 | 88.3 |
| Widowed/Divorced/Separated | 126.1 | 256.6 | 282.5 | 240.9 |
| **Place of residence** | | | | |
| Rural | 56.5 | 90.8 | 95 | 71.6 |
| Urban | 55.2 | 105.8 | 130.4 | 97.8 |
| **Caste/tribe** | | | | |
| Scheduled tribe | 43.2 | 58.4 | 70.3 | 51.6 |
| Scheduled caste | 50.9 | 90 | 98.2 | 71.8 |
| Other backward class | NA | 91.3 | 106 | 75 |
| Others | 58.4 | 111.5 | 121.9 | 101.7 |
| **Religion** | | | | |
| Hindu | NA | 91 | 102.8 | 76.2 |
| Muslim | NA | 103 | 102.2 | 84.7 |
| Christian | NA | 165.1 | 198.9 | 132.4 |
| Sikh | NA | 125.7 | 147.3 | 112.3 |
| Others | NA | 81.5 | 96 | 90.8 |
| **MPCE Wealth Indicator** | | | | |
| Poorest | 42.9 | 66.5 | 68.1 | 52.1 |
| Poorer | 50.9 | 84 | 88.9 | 64.8 |
| Middle | 55.6 | 93.1 | 96.4 | 77.8 |
| Richer | 65.4 | 113.4 | 125.2 | 99.6 |
| Richest | 76.1 | 138.4 | 168.8 | 125.8 |
| **Region** | | | | |
| North | 64.5 | 97.3 | 81.3 | 75.9 |
| Central-East | 48.9 | 78.7 | 94.2 | 72 |
| North-east | 75.6 | 78.1 | 34.2 | 23.6 |
| West | 43.7 | 90 | 80.6 | 73.5 |
| South | 63.9 | 119.7 | 182.8 | 109.3 |
| **India** | **56.2** | **94.7** | **105.6** | **79.3** |

**Table 3. Prevalence shift of major types of self-reported morbidities (per 1000) across socio-demographic determinates in India, 1995–2018.**

| Attributes | In&CD | | | | NCDs | | | | Disability | | | | Injury and other | | | |
|---|---|---|---|---|---|---|---|---|---|---|---|---|---|---|---|---|
| | 1995 | 2004 | 2014 | 2018 | 1995 | 2004 | 2014 | 2018 | 1995 | 2004 | 2014 | 2018 | 1995 | 2004 | 2014 | 2018 |
| **Age Group** | | | | | | | | | | | | | | | | |
| 0–14 | 7.8 | 18.7 | 30.8 | 22.9 | 4.1 | 12.2 | 4.4 | 1.8 | 3.2 | 3.6 | 3.8 | 2 | 33.6 | 39.2 | 32.8 | 31.8 |
| 15–44 | 6.2 | 12.3 | 16.6 | 10.5 | 5.9 | 16.3 | 15.3 | 9.3 | 4.6 | 7.7 | 13.4 | 6.3 | 26.0 | 25.7 | 21 | 17.1 |
| 45–59 | 9.3 | 21.8 | 24.1 | 12.6 | 15.5 | 51.8 | 80.8 | 67.6 | 11.0 | 20.4 | 38.2 | 21.4 | 36.9 | 38.1 | 30.3 | 21.3 |
| 60 and above | 20.3 | 33.7 | 30.8 | 21.5 | 45.3 | 159.1 | 183.2 | 194.6 | 47.1 | 89 | 80.2 | 56.3 | 66.1 | 58.5 | 43 | 31.8 |
| **Sex** | | | | | | | | | | | | | | | | |
| Male | 8.2 | 16.6 | 21.8 | 14.2 | 7.8 | 27.8 | 31 | 26.8 | 6.8 | 12.9 | 14.3 | 8.8 | 31.1 | 31.6 | 25.7 | 20.7 |
| Female | 7.7 | 17.9 | 24.2 | 15.7 | 9.5 | 31.2 | 40.2 | 34.2 | 7.7 | 14.5 | 25.4 | 13.9 | 33.7 | 37.1 | 29.5 | 24.9 |
| **Education** | | | | | | | | | | | | | | | | |
| Illiterate | 9.7 | 22.9 | 31 | 21 | 8.9 | 29.8 | 36.2 | 36.8 | 8.7 | 17.1 | 27.5 | 18.7 | 36.5 | 42.4 | 34.8 | 30.3 |
| Literate and below primary | 6.9 | 14.8 | 21.8 | 18.8 | 7.4 | 21.7 | 29.2 | 22.1 | 5.9 | 9.9 | 15.9 | 7.7 | 30.1 | 30.5 | 28.7 | 27.1 |
| Primary and middle | 5.6 | 12.1 | 21.5 | 12.9 | 8.5 | 29.3 | 35.8 | 30.1 | 6 | 11 | 19.3 | 11.5 | 28.9 | 27.3 | 26.3 | 21.4 |
| Secondary and Higher Secondary | 5.4 | 12.9 | 17.9 | 10.8 | 9.4 | 31.4 | 34.8 | 27.2 | 5.2 | 12.3 | 15 | 8.1 | 24.1 | 27.9 | 22.4 | 17.9 |
| Graduation and above | 11 | 10.1 | 15.3 | 8.6 | 10.9 | 43.2 | 48.1 | 39.7 | 0 | 10 | 14.9 | 8.1 | 69.5 | 22.3 | 18 | 13.8 |
| other | 4.4 | 14.7 | 14.6 | 14.9 | 13.7 | 80.2 | 50.1 | 38.6 | 6.7 | 11.9 | 15.5 | 8.5 | 22 | 39 | 19.6 | 13.5 |
| **Marital status** | | | | | | | | | | | | | | | | |
| Never married | 7 | 15.8 | 25 | 18.4 | 4.2 | 12.3 | 5.3 | 2.9 | 3.4 | 4.6 | 6 | 2.8 | 30.7 | 33.9 | 27.2 | 25.7 |
| Currently married | 8.4 | 17.1 | 20.2 | 11.1 | 11.2 | 39 | 52 | 43.9 | 8.8 | 17.2 | 26 | 14.5 | 31.7 | 31.9 | 26.8 | 18.8 |
| Widowed/Divorced/Separated | 12.8 | 31 | 29.9 | 22.3 | 28.5 | 103.7 | 138.2 | 131.8 | 31 | 65.1 | 77 | 51.2 | 53.8 | 56.8 | 37.4 | 35.6 |
| **Place of residence** | | | | | | | | | | | | | | | | |
| Rural | 8.3 | 18.2 | 22.7 | 15.1 | 7.8 | 24.9 | 26.1 | 23.1 | 7.4 | 13.2 | 19.2 | 10.8 | 33 | 34.5 | 27 | 22.6 |
| Urban | 6.7 | 14.3 | 23.4 | 14.6 | 11.2 | 42.9 | 57.2 | 47.9 | 6.9 | 15 | 20.9 | 12.2 | 30.4 | 33.6 | 28.9 | 23.1 |
| **Caste/tribe** | | | | | | | | | | | | | | | | |
| Scheduled tribe | 6.3 | 14.4 | 24.6 | 14.4 | 4.4 | 12.6 | 11.8 | 8.8 | 3.8 | 7.7 | 12.6 | 9.4 | 28.7 | 23.7 | 21.3 | 19 |
| Scheduled caste | 8.1 | 18.8 | 21.9 | 15.6 | 6.9 | 23.4 | 28 | 23.3 | 6.8 | 12.2 | 19.8 | 10.8 | 32.3 | 35.6 | 28.5 | 22.1 |
| Other backward class | NA | 15.9 | 22.4 | 14.1 | NA | 26.7 | 36.7 | 27.8 | NA | 13.2 | 18.9 | 10.4 | NA | 35.5 | 28 | 22.7 |
| Others | 8.1 | 18.7 | 24 | 16 | 9.6 | 41.3 | 46.4 | 47.4 | 7.8 | 16.9 | 23.3 | 13.8 | 32.9 | 34.6 | 28.2 | 24.5 |
| **Religion** | | | | | | | | | | | | | | | | |
| Hindu | NA | 16.7 | 23.2 | 14 | NA | 27.5 | 33.9 | 29 | NA | 13.3 | 19.3 | 10.9 | NA | 33.5 | 26.4 | 22.3 |
| Muslim | NA | 20.1 | 20.4 | 17.5 | NA | 31.2 | 32.3 | 30.2 | NA | 13 | 19.2 | 11.7 | NA | 38.7 | 30.3 | 25.3 |
| Christian | NA | 18 | 29.1 | 15.9 | NA | 73.8 | 95.5 | 76.3 | NA | 28.4 | 29.4 | 18.1 | NA | 44.9 | 44.9 | 22.1 |
| Sikh | NA | 22.1 | 23.8 | 36.4 | NA | 51.5 | 55 | 35 | NA | 20.4 | 28.9 | 15.8 | NA | 31.7 | 39.6 | 25.1 |
| Others | NA | 14.4 | 24.7 | 19.1 | NA | 32.5 | 32.4 | 35.2 | NA | 13.1 | 19.6 | 14 | NA | 21.5 | 19.3 | 22.5 |
| **MPCE Wealth Indicator** | | | | | | | | | | | | | | | | |
| Poorest | 6.7 | 15.2 | 18.9 | 13.6 | 4.9 | 13.9 | 12.3 | 10.7 | 5 | 8.8 | 13 | 6.8 | 26.3 | 28.6 | 23.9 | 21 |
| Poorer | 7.4 | 17.7 | 24.7 | 13.4 | 6.1 | 19.2 | 20.4 | 17.9 | 6.3 | 12.2 | 19.1 | 9.5 | 31.1 | 34.9 | 24.7 | 24 |
| Middle | 8.6 | 17.7 | 21.6 | 16.4 | 7 | 26.1 | 29.2 | 26.7 | 7.6 | 13.6 | 17.3 | 11.9 | 32.4 | 35.7 | 28.3 | 22.8 |
| Richer | 9 | 19.5 | 24.2 | 16.2 | 10.8 | 39 | 44 | 44.7 | 8.3 | 17.2 | 26.3 | 14.8 | 37.3 | 37.7 | 30.7 | 23.9 |
| Richest | 8.7 | 17 | 27 | 16.2 | 18.2 | 63.9 | 84 | 70.2 | 10.8 | 20.3 | 25.9 | 16.8 | 38.4 | 37.2 | 31.9 | 22.6 |
| **Region** | | | | | | | | | | | | | | | | |
| North | 11 | 21.8 | 21 | 18.9 | 10 | 26.1 | 19.3 | 17.2 | 6.5 | 11.9 | 16.1 | 9.5 | 37 | 37.5 | 24.9 | 30.3 |
| Central-East | 7 | 17.5 | 23.5 | 14.2 | 6.8 | 20.3 | 21.9 | 24.3 | 5.8 | 9.3 | 18.5 | 10.9 | 29.3 | 31.6 | 30.3 | 22.6 |
| North-east | 18.3 | 26.7 | 11.3 | 8.5 | 7.5 | 15.6 | 3.4 | 4.7 | 5.6 | 7.7 | 6.1 | 3.7 | 44.2 | 28.1 | 13.4 | 6.7 |
| West | 6 | 15.4 | 26.6 | 17.4 | 6.8 | 31 | 24.9 | 28.5 | 5.9 | 16.3 | 13.8 | 8.7 | 25 | 27.3 | 15.3 | 18.9 |
| South | 6 | 11.9 | 23.1 | 10.3 | 11 | 46.3 | 86.8 | 60.7 | 11.2 | 20 | 33 | 17.6 | 35.7 | 41.5 | 39.9 | 20.7 |

*(Continued)*

**Table 3.** (Continued)

| Attributes | In&CD | | | | NCDs | | | | Disability | | | | Injury and other | | | |
|---|---|---|---|---|---|---|---|---|---|---|---|---|---|---|---|---|
| | 1995 | 2004 | 2014 | 2018 | 1995 | 2004 | 2014 | 2018 | 1995 | 2004 | 2014 | 2018 | 1995 | 2004 | 2014 | 2018 |
| India | 7.9 | 17.2 | 22.9 | 14.9 | 8.6 | 29.5 | 35.5 | 30.4 | 7.3 | 13.7 | 19.7 | 11.3 | 32.4 | 34.3 | 27.5 | 22.7 |

NA: Data not available for the respective reference period.

Social cast and religion categories have been likewise prevalent over all study period time points. Region-wise morbidity pattern and burden from 1995 to 2018 portrays variations and growth among North, South, Northeast, Western, and Central-East. Recent NSS0-75th rounds data revealed that In&CDs are more dominant in the North region (19 per 1000) than in other regions. However, NCDs (61 per 1000) and disability (18 per 1000) are more prevalent in the South region.

### Temporal changes in the likelihood of all morbidities

Table 4, shows how all morbidity odds change over the period after controlling the effect of socio-demographic and economic determinants. As per the collated all morbidities among different study periods (1995, 2004, 2014 and 2018), the odds for morbidities reported significantly higher over the succeeding decadal period (AOR: 1.81; 95% CI: 1.78, 1.84) times in 2004, (AOR: 2.16; 95% CI: 2.12, 2.2) times in 2014 and (AOR: 1.44; 95% CI: 1.41, 1.46) times in 2018 compared to morbidities in 1995. It also revealed that the highest odds were noted in 2014 but slightly declined in the recent time 2018. In the controlling factors, age showed very high odds for all morbidities in the latter ages of life (AOR: 1.6) in 45–59 and (AOR: 4.1) times in 60 and above compared to the younger age group (0–14), while less likelihood reported among adults (15–44). The urban sector resulted in marginally less likelihood for all morbidities (AOR: 0.94) than the rural sector over the study period.

The MPCE quintile remarkably revealed all morbidities were significantly more prone to the richest (AOR: 2.23) profile people followed by rich (AOR: 1.7), middle (AOR: 2.23), poor (AOR: 1.23) in comparison to the reference of Poorest people. Similarly, other factors such as education, marital status, caste, and region were significant determinants that varied among their counterparts.

### Relative Risk Ratio (RRR) transition among major types of morbidities

Table 5 shows that the risk of In&CDs, NCDs and Disability has been found to be higher in 2004, 2014, and 2018 compared to morbidities in 1995. Specifically, RRR revealed that the risk of In&CDs was more than doubled after a decade in 2004 by 2.26 (95% CI: 2.17, 2.36) times. In 2014, it spiked to triple (RRR = 3.3, 95%CI: 3.17, 3.44), and further, it was slightly declined but still high and almost doubled in 2018 (RRR = 1.94, 95%CI: 1.86, 2.02) compared to In&CDs in 1995. On the same line, reporting NCDs risk has been highly significant (P-value <0.000) higher and more than three times overall time, i.e. (RRR = 3.94, 95%CI: 3.80, 4.08) in 2004, (RRR = 4.74, 95%CI: 4.58, 4.92) in 2014, (RRR = 3.64, 95%CI: 3.51, 3.77) in comparison to NCDs in 1995. Among all morbidities, NCDs' RRR was likely much higher compared to other morbidities across the study period.

The risk of disability was found considerably higher likelihood (RRR) by 2.15, 3.22 and 1.62 times in 2004, 2014 and 2018, respectively, compared to the disabilities in the reference year 1995. It showed that the risk of Disabilities suffering over time was lower compared to the In&CDs and NCDs. Although In&CDs, NCDs, and Disability have higher RRR over the

**Table 4. Adjusted odds ratio (AOR) for all morbidities through pooled multiple regression analysis according to the socio-demographic characteristics, India.**

| Attributes | AOR | 95% CI | |
|---|---|---|---|
| **NSSO round (year)** | | | |
| 52 (1995–96) | 1.00 | - | - |
| 60 (2004) | 1.81 | 1.78 | 1.84 |
| 71 (2014) | 2.16 | 2.12 | 2.20 |
| 75 (2018) | 1.44 | 1.41 | 1.46 |
| **Age** | | | |
| 0–14 | 1.00 | - | - |
| 15–44 | 0.67 | 0.66 | 0.69 |
| 45–59 | 1.58 | 1.54 | 1.63 |
| 60+ | 4.06 | 3.95 | 4.18 |
| **sex** | | | |
| Male | 1.00 | | |
| Female | 1.06 | 1.05 | 1.07 |
| **Education level** | | | |
| Illiterate | 1.00 | - | - |
| Literate and below primary | 0.84 | 0.83 | 0.86 |
| Primary and middle | 0.87 | 0.85 | 0.89 |
| Secondary and Higher Secondary | 0.78 | 0.77 | 0.79 |
| Graduation and above | 0.63 | 0.61 | 0.65 |
| Diploma and other | 0.67 | 0.64 | 0.70 |
| **Marital Status** | | | |
| Never married | 1.00 | - | - |
| currently married | 1.34 | 1.31 | 1.37 |
| Widowed/divorced/ separated | 1.54 | 1.49 | 1.58 |
| **Sector** | | | |
| Rural | 1.00 | | |
| Urban | 0.94 | 0.93 | 0.95 |
| **MPCE Quintile** | | | |
| Poorest | 1.00 | - | - |
| Poor | 1.23 | 1.21 | 1.26 |
| Middle | 1.42 | 1.40 | 1.45 |
| Rich | 1.70 | 1.67 | 1.74 |
| Richest | 2.32 | 2.28 | 2.37 |
| **Caste** | | | |
| ST | 1.00 | - | - |
| SC | 1.37 | 1.34 | 1.41 |
| OBC | 1.27 | 1.24 | 1.30 |
| Others | 1.38 | 1.35 | 1.41 |
| **Regional Zone** | | | |
| North | 1.00 | - | - |
| Central | 1.11 | 1.10 | 1.13 |
| East | 1.19 | 1.16 | 1.23 |
| North-east | 0.58 | 0.57 | 0.60 |
| West | 0.94 | 0.92 | 0.95 |
| South | 1.60 | 1.58 | 1.63 |
| **Modal Other Parameters** | **Constant:** 0.0384085 | **Std. Error:** 0.0083 | **P-value:** 0.0000 |
| | **Pseudo R²:** 0.1061 | **Observation:** 1909941 | **Prob > chi2:** 0.0000 |

**Table 5. Multinomial regression analysis for morbidity risk transition over the three decadal period in India, 1995–2018.**

| Comparison outcome | Infectious/CDs | | | NCDs | | | Disability | | | Injury and other | | |
|---|---|---|---|---|---|---|---|---|---|---|---|---|
| Base outcome | NO | | | NO | | | NO | | | NO | | |
| Attributes | RRR | 95% CI | | RRR | 95% CI | | RRR | 95% CI | | RRR | 95% CI | |
| **NSSO Round (Year)** | | | | | | | | | | | | |
| 52nd (1995) | 1.00 | - | - | 1.00 | - | - | 1.00 | - | - | 1.00 | - | - |
| 60th (2004) | 2.26 | 2.17 | 2.36 | 3.94 | 3.80 | 4.08 | 2.15 | 2.05 | 2.25 | 1.04 | 1.01 | 1.07 |
| 71st (2014) | 3.30 | 3.17 | 3.44 | 4.74 | 4.58 | 4.92 | 3.22 | 3.08 | 3.37 | 0.95 | 0.92 | 0.97 |
| 75th (2018) | 1.94 | 1.86 | 2.02 | 3.64 | 3.51 | 3.77 | 1.62 | 1.55 | 1.70 | 0.65 | 0.63 | 0.67 |
| **Age** | | | | | | | | | | | | |
| 0–14 | 1.00 | - | - | 1.00 | - | - | 1.00 | - | - | 1.00 | - | - |
| 15–44 | 0.58 | 0.55 | 0.61 | 0.95 | 0.90 | 1.00 | 2.28 | 2.13 | 2.44 | 0.66 | 0.64 | 0.68 |
| 45–59 | 0.81 | 0.76 | 0.86 | 4.15 | 3.91 | 4.41 | 5.84 | 5.42 | 6.29 | 0.88 | 0.84 | 0.92 |
| 60+ | 1.48 | 1.39 | 1.58 | 12.47 | 11.74 | 13.24 | 17.79 | 16.53 | 19.15 | 1.52 | 1.45 | 1.59 |
| **sex** | | | | | | | | | | | | |
| Male | 1.00 | - | - | 1.00 | - | - | 1.00 | - | - | 1.00 | - | - |
| Female | 0.93 | 0.90 | 0.95 | 1.10 | 1.08 | 1.12 | 1.11 | 1.08 | 1.15 | 1.07 | 1.05 | 1.09 |
| **Education level** | | | | | | | | | | | | |
| Illiterate | 1.00 | - | - | 1.00 | - | - | 1.00 | - | - | 1.00 | - | - |
| Literate and below primary | 0.65 | 0.63 | 0.67 | 1.12 | 1.09 | 1.16 | 0.98 | 0.94 | 1.02 | 0.74 | 0.73 | 0.76 |
| Primary and middle | 0.59 | 0.57 | 0.62 | 1.18 | 1.14 | 1.22 | 0.91 | 0.87 | 0.95 | 0.76 | 0.73 | 0.78 |
| Secondary and higher Secondary | 0.55 | 0.53 | 0.57 | 1.05 | 1.02 | 1.08 | 0.74 | 0.71 | 0.77 | 0.65 | 0.63 | 0.67 |
| Graduation and above | 0.41 | 0.38 | 0.44 | 0.87 | 0.84 | 0.91 | 0.46 | 0.43 | 0.50 | 0.48 | 0.45 | 0.51 |
| Diploma and other | 0.51 | 0.45 | 0.58 | 0.98 | 0.91 | 1.05 | 0.65 | 0.58 | 0.72 | 0.54 | 0.50 | 0.59 |
| **Marital Status** | | | | | | | | | | | | |
| Never married | 1.00 | - | - | 1.00 | - | - | 1.00 | - | - | 1.00 | - | - |
| currently married | 1.09 | 1.04 | 1.14 | 2.14 | 2.04 | 2.24 | 1.16 | 1.10 | 1.22 | 1.11 | 1.07 | 1.14 |
| Widowed/divorced/ separated | 1.25 | 1.16 | 1.34 | 2.42 | 2.29 | 2.55 | 1.57 | 1.47 | 1.67 | 1.28 | 1.21 | 1.34 |
| **Place of Residence** | | | | | | | | | | | | |
| Rural | 1.00 | - | - | 1.00 | - | - | 1.00 | - | - | 1.00 | - | - |
| Urban | 0.87 | 0.85 | 0.90 | 1.08 | 1.06 | 1.11 | 0.86 | 0.84 | 0.89 | 0.88 | 0.86 | 0.90 |
| **MPCE Quintile** | | | | | | | | | | | | |
| Poorest | 1.00 | - | - | 1.00 | - | - | 1.00 | - | - | 1.00 | - | - |
| Poor | 1.24 | 1.19 | 1.29 | 1.38 | 1.32 | 1.43 | 1.21 | 1.15 | 1.27 | 1.20 | 1.16 | 1.23 |
| Middle | 1.37 | 1.32 | 1.43 | 1.70 | 1.64 | 1.77 | 1.36 | 1.30 | 1.43 | 1.38 | 1.34 | 1.42 |
| Rich | 1.53 | 1.47 | 1.59 | 2.29 | 2.21 | 2.38 | 1.61 | 1.53 | 1.68 | 1.54 | 1.49 | 1.58 |
| Richest | 1.90 | 1.82 | 1.99 | 3.46 | 3.34 | 3.60 | 2.06 | 1.96 | 2.17 | 1.88 | 1.82 | 1.94 |
| **Caste** | | | | | | | | | | | | |
| ST | 1.00 | - | - | 1.00 | - | - | 1.00 | - | - | 1.00 | - | - |
| SC | 1.21 | 1.15 | 1.27 | 1.60 | 1.52 | 1.68 | 1.31 | 1.23 | 1.39 | 1.41 | 1.36 | 1.47 |
| OBC | 1.08 | 1.03 | 1.13 | 1.48 | 1.41 | 1.55 | 1.20 | 1.13 | 1.27 | 1.36 | 1.31 | 1.41 |
| Others | 1.14 | 1.09 | 1.19 | 1.77 | 1.69 | 1.86 | 1.36 | 1.28 | 1.44 | 1.33 | 1.28 | 1.37 |
| **Regional Zone** | | | | | | | | | | | | |
| North | 1.00 | - | - | 1.00 | - | - | 1.00 | - | - | 1.00 | - | - |
| Central | 1.05 | 1.02 | 1.09 | 1.20 | 1.16 | 1.23 | 1.01 | 0.97 | 1.06 | 1.09 | 1.06 | 1.12 |

*(Continued)*

**Table 5.** (Continued)

| Comparison outcome | Infectious/CDs | | | NCDs | | | Disability | | | Injury and other | | |
|---|---|---|---|---|---|---|---|---|---|---|---|---|
| **Base outcome** | **NO** | | | **NO** | | | **NO** | | | **NO** | | |
| **Attributes** | **RRR** | **95% CI** | | **RRR** | **95% CI** | | **RRR** | **95% CI** | | **RRR** | **95% CI** | |
| East | 1.02 | 0.95 | 1.10 | 1.53 | 1.42 | 1.64 | 1.09 | 1.00 | 1.18 | 1.02 | 0.98 | 1.07 |
| North-east | 0.78 | 0.74 | 0.82 | 0.43 | 0.40 | 0.45 | 0.40 | 0.37 | 0.43 | 0.65 | 0.62 | 0.67 |
| West | 0.95 | 0.91 | 0.99 | 1.14 | 1.11 | 1.18 | 0.91 | 0.87 | 0.95 | 0.77 | 0.74 | 0.79 |
| South | 0.94 | 0.91 | 0.98 | 2.38 | 2.32 | 2.44 | 1.65 | 1.59 | 1.72 | 1.30 | 1.27 | 1.34 |
| Other parameter as per Disease category: | Constant: .0221 | Standard Error: 0.0092 | P-value: 0.000 | Constant: .00048 | Standard Error: 0.00034 | P-value: 0.000 | Constant: .00116 | Standard Error: 0.00083 | P-value: 0.000 | Constant: .04278 | Standard Error: 0.0112 | P-value: 0.000 |
| Overall model parameter | **Pseudo R$^2$**: 0.1178 | | | **Prob > chi2**: 0.0000 | | | **Log likelihood**: (-653739) | | | **Observation**: 1909941 | | |

Note: The above results were based on a single Multiple Multinomial Regression mode. RRR: Relative Risk ratio.

period, Injury and Other diseases have shown reverse results; it has been estimated to be significantly lower RRR in the same period. In 2004 risk of Injury and Other diseases (RRR = 1.04, 95%CI: 1.01, 1.07) were almost the same as in the reference year 1995 and then reduced to (RRR = 0.95, 95%CI: 0.92, 0.97) in 2014 and (RRR = 0.65, 95%CI: 0.63, 0.67) in 2018. Further, other models controlling socio-demographic factors such as age, sex, education, marital status, urban-rural differential, caste and region were found to be significant determinants, revealing the distinguished relative risk ratio within their categories and among the different morbidities in Table 5.

## Discussion

The present research delves into the nuanced evolution of morbidity patterns over contemporary and preceding decades. This study aimed to document the progressing morbidity patterns based on self-reported morbidities over (1995–2018) while uncovering the narratives driving these transitions. This dynamic transition is intricately linked to demographic shifts, elucidating the overarching concept of epidemiologic transition. Our study ventures beyond mere documentation of temporal changes; it endeavours to formulate hypotheses delineating the transformative trajectory of disease burdens. In the recent past, many studies have discussed the epidemiological transition based on various aspects using demographic and morbidity data in India [9,11,18,19]. However, our study contributes additional insights by extending beyond a national perspective and conducting a meticulous analysis at the state level and determinants, especially in morbidity context. This in-depth exploration includes an examination of determinants influencing these shifts in morbidity patterns. It aligns with previous researchers' recommendation that similar exercises should be repeated over temporal periods for enhanced policy and program development inputs in public health research [20].

Our study demonstrated that the prevalence of various morbidities has exhibited a consistent and doubling trend between 1995 and 2014 as similarly observed by existing literature [9] in India. However, this study delves deeper into subsequent data, revealing a noteworthy decline in that trend in 2018. It showed the parabolic nature of the pattern of diseases over three decades. A parallel scenario is observed in the realm of health, with Infectious and Communicable Diseases (In&CDs), Non-communicable diseases (NCDs), and disability, excluding Injuries, experiencing a decline since 2004. Despite a modest decrease in prevalence, the period from 1995 to 2018 witnessed a threefold increase in Non-communicable Diseases (NCDs) and

nearly a twofold rise in Infectious and Communicable Diseases (In&CDs). In contrast, injuries and other conditions exhibited a decline during the same period. This pattern reveals that India continues to grapple with the dual burden of diseases (NCDs and In&CDs), exhibiting a distinctive pattern aligned with the epidemiological transition seen in many developing countries, previously noted by another study [19]. However, NCDs' burden escalates faster than In&CDs, with this disparity pronounced at both early and later stages of life, which was consistent finding with another study [21,22].

Furthermore, a crucial aspect of this study is its examination of morbidity transition at the sub-national level. Interestingly, the findings reveal extensive variation across Indian regions and states, showcasing a wide range of prevalence rates. This study shows that the southern region has the most disease burden reported, specifically of NCDs and Disability, whereas In&CDs were most prevalent in the northern and western parts of India, which has argued on the same line by the other authors [23,24]. However, this study further added the north-eastern region reported the lowest burden from 1995 to 2018. Likewise, in a more detailed state-wise analysis, Kerala consistently stands out with the highest reported morbidities, with states like Lakshadweep, Andhra Pradesh, West Bengal, and Punjab also displaying noteworthy prevalence rates. The elevated prevalence of self-reported morbidities in Kerala could be attributed to several factors, including a higher proportion of the elderly population, as well as superior socio-economic status and educational literacy compared to other states [25,26].

In contrast, the north-eastern states, particularly Manipur, exhibit the lowest prevalence of morbidity conditions, followed by Mizoram, Nagaland, Meghalaya, and other north-eastern states, which also follow a similar trend of relatively low morbidity rates addressed by other regional study [27]. It could be due to a substantially lower reporting of NCDs disabilities, and injuries in the northeastern states. However, there was a higher reporting burden for In&CDs over the same period [9,28]. Similarly, the state-level burden pattern of disability was reported in parabolic transition over three decades, which showed declining disability in the contemporary period. As with NCDs burden, Kerala has reported the highest burden of disability and Injury and Other diseases amongst the Indian states. There were several other states, West Bengal, Chandigarh, Orissa, Andhra Pradesh, Haryana, Uttar Pradesh, Dadara & Nagar Haveli and Delhi, etc., have a higher burden of Injury and Other diseases than the national average.

This study emphasised the significant role of self-reported morbidities in explaining variations across diverse demographic and socio-economic strata in India from 1995 to 2018. Amongst all demographic indicators, age plays a vital role in the severity and burden of diseases, from a higher burden of Infection and communicable at the earliest ages of life to NCDs at the latter ages of life, earlier studies also show the comparable importance of demographic factors necessary to address timely the most prevalent section of population [18,21,29]. Notably, our study observed, that individuals aged 60 and above emerge as the most vulnerable, consistently exhibiting a higher prevalence of various morbidities. This finding aligns with other studies, emphasising the importance of addressing health concerns in this age group [23,30].

Likewise, sex also plays an important role in determining the encumbrance of morbidities over the study period; In&CDs, NCDs, Disability and Injuries were considerably more predominant among females than males alike scenarios observed by existing work in different settings of the population [18,31]. It could be linked to a longer life expectancy among women than men. Present and previous study hypotheses found consistent results as individuals age, chronic conditions tend to increase, resulting in a morbidity expansion scenario in India [32]. In the same way, other background factors, self-reported morbidities, were well determined by the strata of education level, marital status, residence, social cast, religion and wealth indicator in India [9].

Now, the central focus of the study was to discuss the morbidity transition and its risk. Under the pooled analysis study observed since 1995, the risk of Injuries and other diseases considerably declined, while the risk of In&CDs, NCDs and Disability has been found to be higher in 2004, 2014, and 2018 compared to morbidities in 1995. However, NCD risk has been consistently very high, but In&CD risk has noticeably declined. Currently, India deals with a dual burden of diseases, encompassing both In&CDs and NCDs [19]. However, on-going socio-demographic changes contribute to the escalating burden of NCDs [11,21] gradually replacing In&CDs at the national and sub-national levels. This aligns with the proposition of epidemiological transition, suggesting a shift in disease patterns over and in the upcoming decades.

## Conclusion

As India undergoes concurrent demographic and epidemiological transitions, our study aligns with this dynamic shift. Notably, it highlights significant disparities in reporting morbidity burdens across different states in India from 1995 to 2018. These variations were attributed to distinct demographic, social, and economic determinants characterising each state, contributing to the nuanced landscape of health challenges across the nation. The ageing population and rising life expectancy are concurrently fuelling the prevalence of NCDs without necessarily displacing existing infectious and communicable morbidities. Among all the morbidities over the period of disease transition from 1995 to 2018, NCDs are most predominantly gaining share.

This study is evident that the southern region has the most reported burdens, of which Kerala consistently stands out with the highest reported morbidity rates, followed by states like Lakshadweep, Andhra Pradesh, Karnataka, West Bengal, Punjab and others. Most of these states have the highest share of NCDs, followed by In&CDs, Disability and injuries. Conversely, the northern eastern states, particularly Manipur, Mizoram, Nagaland, and Meghalaya, exhibit the lowest prevalence of morbidity conditions over the study period. In the broader state-level context, the pattern of morbidity conditions appears to resemble a parabolic curve. It started with relatively low levels in 1995 and then exhibited an upward trend in 2004 and 2014 before gradually declining after that up to 2018, but still at a higher level as per recent prevalence, especially for NCDs. A nuanced comprehension of the burden of morbidities can be attained by examining socio-demographic determinants. Factors such as age, sex, residence, education, caste, religion, and wealth all play crucial roles in elucidating the severity of various disease burdens within different population segments.

### Policy recommendations

With these findings, the study recommends integrated and decentralised health policies to address NCDs, In&CDs, disabilities, and injuries at both the national and state levels. These policies should be tailored to the specific disease burdens in each region, ensuring a comprehensive and targeted approach to healthcare. There is a pressing need for health education and awareness initiatives to promote the significance of a healthy lifestyle, preventive measures, and early detection. These efforts should be strategically designed to reach diverse demographic groups. Additionally, in a populous country like India, characterised by diverse regions, there is a critical requirement for health infrastructure development, research and innovation, community engagement, as well as financial support and insurance to address the multifaceted health challenges effectively.

### Strength and limitation

In the current landscape of recent literature, the present study is unique in its comprehensive portrayal of the morbidity burden and its progression spanning from 1994 to 2018. This

distinctive contribution is based on an analysis of four rounds of NSSO data, encompassing both the national and sub-national levels. Also, this study provides the determinants through India's socio-demographic and economic aspects. This research uniquely delves into the determinants of morbidity, exploring socio-demographic and economic aspects. However, it is essential to acknowledge certain limitations. Given the cross-sectional nature of the data, the study primarily establishes associations rather than delving into causal analyses. Furthermore, the reliance on self-reported morbidities introduces the possibility of underestimation or overestimation of prevalence, influenced by various factors. This study relies on secondary data analysis, which means there may be an inherent flaw in how age groups and education statuses were represented. This could result in variations in proportional distribution over study period. As a result, comparing study participants across different time frames and developmental stages may impact their awareness levels, thus this may somehow affect the overall comparison process. Lastly, the study does not incorporate other potential risk factors like physical activity, drinking or smoking behaviours, or nutrition. Recognising these limitations opens avenues for further research and the identification of gaps in the existing understanding of morbidity dynamics.

## Acknowledgments

We acknowledge National Sample Survey Office (NSSO) under the Ministry of Statistics and Programme Implementation (MOSPI) for accessing the relevant data for this study.

## Author Contributions

**Conceptualization:** Mahadev Bramhankar, Murali Dhar.

**Data curation:** Mahadev Bramhankar.

**Formal analysis:** Mahadev Bramhankar.

**Investigation:** Murali Dhar.

**Methodology:** Mahadev Bramhankar.

**Supervision:** Murali Dhar.

**Validation:** Murali Dhar.

**Visualization:** Mahadev Bramhankar.

**Writing – original draft:** Mahadev Bramhankar.

**Writing – review & editing:** Mahadev Bramhankar, Murali Dhar.

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
