## [Decision Letter · Decision Letter 0]

8 May 2024

PONE-D-24-11744Morbidity transition at the national and sub-national level and their determinants over the past and contemporary period in India

PLOS ONE

Dear Dr. Bramhankar,

Thank you for submitting your manuscript to PLOS ONE. After careful consideration, we feel that it has merit but does not fully meet PLOS ONE’s publication criteria as it currently stands. Therefore, we invite you to submit a revised version of the manuscript that addresses the points raised during the review process.

Reviewer 1

1TpBetter to use the full form when it first appears in the text like NSSO, ID etc. in the abstract.2. Table 2: Mention years in age group3. Table 2: Education: Middle school is taken in two consecutive rows4. Table 4: Middle school is omitted here5. Better to provide the complete output of logistic regression model like model fitness, variance explained by independent variables etc.6. Whether all the disease entities were included in each NSSO surveys starting from 1995 till 2018? Otherwise, the diseases that were not recorded in the earlier surveys will come in a big way in the next subsequent surveys. It needs to be discussed in discussion section. Moreover, please mention throughout the surveys the operational definition of the disease category also remained same.7. Mention the name of the statistical software used in the study8. Objective: "........exploring the socio-demographic factors that play a crucial role in shaping these patterns over the past three decades" - better not to use the action verb 'explore' in this study. Reviewer 2:Among limitations it may be added that being a secondary data analysis, an inherent flaw of the representation of the age group wise and education status wise may not be exactly similar in proportional distribution. 

We look forward to receiving your revised manuscript.

Kind regards,

Arun Kumar Sharma

Academic Editor

PLOS ONE

2. Please include a separate caption for each figure in your manuscript.

Additional Editor Comments:

Authors may please revise as suggested by the Reviewers.

Reviewers' comments:

Reviewer's Responses to Questions

**Comments to the Author**

1. Is the manuscript technically sound, and do the data support the conclusions?

Reviewer #1: Yes

Reviewer #2: Yes

2. Has the statistical analysis been performed appropriately and rigorously? 

Reviewer #1: Yes

Reviewer #2: Yes

3. Have the authors made all data underlying the findings in their manuscript fully available?

Reviewer #1: Yes

Reviewer #2: Yes

4. Is the manuscript presented in an intelligible fashion and written in standard English?

Reviewer #1: Yes

Reviewer #2: Yes

5. Review Comments to the Author

Reviewer #1: The manuscript needs few clarifications:

The manuscript is nicely written.

1. Better to use the full form when it first appears in the text like NSSO, ID etc. in the abstract.

2. Table 2: Mention years in age group

3. Table 2: Education: Middle school is taken in two consecutive rows

4. Table 4: Middle school is omitted here

5. Better to provide the complete output of logistic regression model like model fitness, variance explained by independent variables etc.

6. Whether all the disease entities were included in each NSSO surveys starting from 1995 till 2018? Otherwise, the diseases that were not recorded in the earlier surveys will come in a big way in the next subsequent surveys. It needs to be discussed in discussion section. Moreover, please mention throughout the surveys the operational definition of the disease category also remained same.

7. Mention the name of the statistical software used in the study

8. Objective: "........exploring the socio-demographic factors that play a crucial role in shaping these patterns over the past three decades" - better not to use the action verb 'explore' in this study.

Reviewer #2: Reviewer comments:

Overall, well written research with due appreciation of limitation and strengths given it’s a cross sectional nature.

Comment:

Among limitations it may be added that being a secondary data analysis, an inherent flaw of the representation of the age group wise and education status wise may not be exactly similar in proportional distribution.

6. PLOS authors have the option to publish the peer review history of their article (what does this mean?). If published, this will include your full peer review and any attached files.

Reviewer #1: **Yes: **Indranil Saha

Reviewer #2: No

---

## [Author Response · Author response to Decision Letter 0]

9 May 2024

Thank you to both the reviewer and handling academic editor for the valuable comments and suggestions for the betterment of our study. We have responded and incorporated in the revised manuscript as entire points raised by the reviewers as given below. 

Editor:

Comment: Authors may please revise as suggested by the Reviewers.

Response: Thank you so much for the handling and assigned the manuscript to better knowledgeable reviewer from the same domain. As per your comment, we have responded all queries and clarification raised by the reviewer. 

Reviewer 1: 

Overall Comment: The manuscript needs few clarifications:

Response: Thank you for your valuable time to reviewing our manuscript. We have responded to all your queries and clarifications as given below.

Query 1: Better to use the full form when it first appears in the text like NSSO, ID etc. in the abstract.

Response: Thank you for your valuable suggestion. We have made the changes as you suggested in the revised manuscript. 

Query 2: Table 2: Mention years in age group.

Response: Thank you for your notice. We have made the changes as you suggested in the revised manuscript. 

Query 3: Table 2: Education: Middle school is taken in two consecutive rows

Response: Thank you for your remark. It was a typo mistake; we have rechecked our coding and analysis, and it was a typo mistake in the tables. We have made changes to the tables accordingly.

Query 4: Table 4: Middle school is omitted here.

Response: It was missed to write the “middle” in the education variable category. The entire variable and subcategory used throughout the analysis were the same for all the respective tables. We have checked and corrected it in the revised manuscript. Thank you for your important remark.

Query 5: Better to provide the complete output of logistic regression model like model fitness, variance explained by independent variables etc.

Response: Thank you for the suggestion. Your suggestion is essential, as the other model parameter should be disclosed with the regression model; however, our table size was already big and lengthy; therefore, we have excluded that part and provided the main result of the model. But as per your suggestion, we have provided all the remaining results and other parameters at the end of the regression table in the revised manuscript.

Query 6: Whether all the disease entities were included in each NSSO surveys starting from 1995 till 2018? Otherwise, the diseases that were not recorded in the earlier surveys will come in a big way in the next subsequent surveys. It needs to be discussed in discussion section. Moreover, please mention throughout the surveys the operational definition of the disease category also remained same.

Response: Thank you for asking an important query. For comparability in all-around NSSO, they have collected and followed almost all standard diseases in each round, which were prevalent in that respective time and other major diseases. All disease entities were nearly similar. We checked all disease categorisation in all four round of NSSO survey and major disease classification was done as per the standard ICD-10 classification. It was a very hectic and mental exercise for us. For the comparisons approach, we have also referred to the NSSO health round comparison guidelines from their reports and earlier published literature, which has already been cited in the manuscript's main text. 

Query 7: Mention the name of the statistical software used in the study.

Response: Thank you for the notice about it. We have used the STATA 16 for statistical analysis purposes. We have mentioned the same in the statistical methodology section.

Query 8: Objective: "........exploring the socio-demographic factors that play a crucial role in shaping these patterns over the past three decades" - better not to use the action verb 'explore' in this study.

Response: Thank you for the comment. You have noticed rightly that for this study “Explore” word will be unsuitable as per the study type. We have changed instead of well suitable word instead of it in the aim and objective part of the manuscript. Thank You!

 

Reviewer 2: 

Overall Comment: Overall, well written research with due appreciation of limitation and strengths given it’s a cross sectional nature.

Response: Thank you for the valuable comment and time for review. We have tried to answer all your comments as given below. 

Comment: 1: Among limitations it may be added that being a secondary data analysis, an inherent flaw of the representation of the age group wise and education status wise may not be exactly similar in proportional distribution. It is of a value to consider that the increasing life expectancy, urbanization and increasing access to social media via digital platforms, might have impacted, the awareness levels among the participants reporting the information, which may differ considerably, over the years from 1995 till 2018. This may, also contribute to an over estimation in the reported responses for morbidity.

Response: Thank you very much for your valuable comment and keen observation. The comment you raise was absolutely in line with our study with transition phenomenon from 1995 to 2018 and at four points of cross sectional data. As a result, through the lens of Age distribution and education, comparing study participants across different time frames and developmental stages may impact their awareness levels, thus, this may somehow affect the overall comparison process. We have incorporated this as a limitation by summarising it in the revised manuscript.

---

## [Editor Report · Decision Letter 1]

14 May 2024

Morbidity transition at the national and sub-national level and their determinants over the past and contemporary period in India

PONE-D-24-11744R1

Dear Dr. Brahmankar,

We’re pleased to inform you that your manuscript has been judged scientifically suitable for publication and will be formally accepted for publication once it meets all outstanding technical requirements.

Kind regards,

Arun Kumar Sharma

Academic Editor

PLOS ONE

Additional Editor Comments (optional):

None
---

## [Editor Report · Acceptance letter]

30 May 2024

PONE-D-24-11744R1 

PLOS ONE

Dear Dr. Bramhankar, 

I'm pleased to inform you that your manuscript has been deemed suitable for publication in PLOS ONE. Congratulations! Your manuscript is now being handed over to our production team.

Kind regards, 

on behalf of

Professor Arun Kumar Sharma 

Academic Editor

PLOS ONE